# Influence of TOPO and TOPO-CdSe/ZnS Quantum Dots on Luminescence Photodynamics of InP/InAsP/InPHeterostructure Nanowires

**DOI:** 10.3390/nano11030640

**Published:** 2021-03-05

**Authors:** Artem I. Khrebtov, Vladimir V. Danilov, Anastasia S. Kulagina, Rodion R. Reznik, Ivan D. Skurlov, Alexander P. Litvin, Farrukh M. Safin, Vladislav O. Gridchin, Dmitriy S. Shevchuk, Stanislav V. Shmakov, Artem N. Yablonskiy, George E. Cirlin

**Affiliations:** 1Alferov University, Khlopinast. 8/3, 194021 St. Petersburg, Russia; a.s.panfutova@gmail.com (A.S.K.); moment92@mail.ru (R.R.R.); dmitrii.shevchuk@gmail.com (D.S.S.); stas-svs@list.ru (S.V.S.); cirlin.beam@mail.ioffe.ru (G.E.C.); 2Emperor Alexander I St. Petersburg State Transport University, Moskovsky pr. 9, 191031 St. Petersburg, Russia; vdanilov039@gmail.com; 3Ioffe Institute, Polytechnicheskayast. 26, 194021 St. Petersburg, Russia; 4ITMO University, Kronverkskiy pr. 49, 197101 St. Petersburg, Russia; ivan.skurlov.23@gmail.com (I.D.S.); litvin@itmo.ru (A.P.L.); farruhsafin@gmail.com (F.M.S.); 5Institute for Analytical Instrumentation RAS, Ivana Chernykhst. 31-33 A, 190103 St. Petersburg, Russia; 6St. Petersburg State University, Ulyanovskayast. 1, Peterhof, 198504 St. Petersburg, Russia; gridchinvo@yandex.ru; 7Institute for Physics of Microstructures RAS, GSP-105, Academicheskayast. 7, Afonino, 603950 Nizhny Novgorod, Russia; yablonsk@ipm.sci-nnov.ru; 8ETU “LETI”, Professora Popova st. 5, 197376 St. Petersburg, Russia

**Keywords:** nanowires, TOPO ligands, molecular-beam epitaxy, luminescence kinetics, reverse transfer

## Abstract

The passivation influence by ligands coverage with trioctylphosphine oxide (TOPO) and TOPO including colloidal CdSe/ZnS quantum dots (QDs) on optical properties of the semiconductor heterostructure, namely an array of InP nanowires (NWs) with InAsP nanoinsertion grown by Au-assisted molecular beam epitaxy on Si (111) substrates, was investigated. A significant dependence of the photoluminescence (PL) dynamics of the InAsP insertions on the ligand type was shown, which was associated with the changes in the excitation translation channels in the heterostructure. This change was caused by a different interaction of the ligand shells with the surface of InP NWs, which led to the formation of different interfacial low-energy states at the NW-ligand boundary, such as surface-localized antibonding orbitals and hybridized states that were energetically close to the radiating state and participate in the transfer of excitation. It was shown that the quenching of excited states associated with the capture of excitation to interfacial low-energy traps was compensated by the increasing role of the “reverse transfer” mechanism. As a result, the effectiveness of TOPO-CdSe/ZnS QDs as a novel surface passivation coating was demonstrated.

## 1. Introduction

The optical, electrical, thermal, and mechanical properties of semiconductor nanowires (NWs) are currently being actively investigated for use in solar cells, photoemitters, detectors, nanogenerators, etc. [1,2]. Among them, large band gap InP NWs containing smaller band gap InAsP insertion attracted an increasing attention due to their applications in infrared photodetection and optical communication [3,4]. The band gap of InAsP can be tailored by adjusting the alloy composition and, in particular, can cover the 1.30–1.55 mkm wavelength range. However, the high surface-to-volume ratio of NWs makes effective surface passivation critical for device applications. Wet chemical treatments and the atomic layer deposition of Al_2_O_3_, AlN, TiN, GaN, and TiO_2_ on InP nanowires have been shown to provide some degree of surface passivation, but this is generally not stable [5]. LaPierre et al. demonstrated the passivation effectiveness of InP NWs by the grown AlInP shell [6]. Vugt et al. showed in [7] that the luminescence efficiency of InP nanowires can be improved by photo-assisted wet chemical etching in a butanol solution containing HF and the indium-coordinating trioctylphosphine oxide (TOPO) ligand.

We have recently shown [8,9] that the deposition of a TOPO quasi-Langmuir layer containing CdSe/ZnS colloidal dots on the arrays of InP/InAsP/InP nanowires leads to a significant increase in the duration and intensity of the photoluminescence of the InAsP quantum dot like nanoinsertion (QI). This work aimed to shed more light onthe study of the luminescence photodynamics of the InP/InAsP/InP NWs ensemble passivated by TOPO ligands and by TOPO-CdSe/ZnS quantum dots (QD) layers under excitation close to the absorbance band of CdSe/ZnS QDs and in its transparent region.

## 2. Materials and Methods

The InP nanowires with InAsP insertions were grown on Si (111) substrate by Au-assisted molecular beam epitaxy on a Compact 21 Riber setup. The growth procedure is described in detail in [10]. The NWs were formed predominantly in the (111) direction and had a wurtzite phase. The resulting morphology of the NWs ensembles and structural properties of the NWs of the obtained hybrid structure were investigated by scanning electron microscopy (SEM) and transmission electron microscopy (TEM) and were presented in previous works [8,10]. The average height of the NWs was 4 μm, and the diameter turned out to be nonuniform in height and amounted to 100 nm at the base and 30 nm at the top of the NWs, with asurface density of about 3 × 10^8^ cm^−2^. As it has been shown by the TEM studies of NWs, the dimensions of the InAsP QI (As content is 40%) were 60 nm in length and 15 nm in diameter, and the shell around the QI was about 10 nm. In addition to the InAsP QI itself, an InAsP quantum well (QW) (the average arsenic content is 15–25%) was formed inside the NW. The QW appeared as a result ofthe lateral growth of a material during the formation of QI and was a radial layer several nanometers thick. Then, either pure TOPO with concentrations of 7 × 10^−7^ mol/L and 1.5 mol/L (with TOPO molar mass equals to 386.6 g/mol) or the solution of colloidal QDs in toluene with a concentration of about 10^−5^ M were applied to substrate with the NWs by the drop-cast. Deposed QDs had a structure with a CdSe core (about 3 nm in diameter) covered with a ZnS shell and TOPO ligand layer. The wavelength of the QD photoluminescence (PL) maximum was 575 nm. The average distance between QDs in the obtained quasi-Langmuir layer was about 5 nm. The resulting hybrid structure was stable in the air and practically did not change the PL parameters over time.

Photoluminescence (PL) signal was measured using custom-built spectrofluorimetres. A Si-based CCD camera AndoriDus 401 (Andor, Belfast, Northern Ireland) was used as a detector and a 532 nm Nd:YAGcw-laser was the laser source used in the 500–1100 nm spectral range. For PL in near IR, we used the InGaAs cooled photodiode Hamamatsu G5852-21 (Hamamatsu Photonics K.K., Hamamatsu, Japan) as a detector and the 633 nm HeNe cw-laser (PLASMA, Ryazan, Russia). Secondary radiation was collected according to the standard scheme at an angle of 90°, and the exciting radiation was cut off by a KS19 light filter (cutoff wavelength was about 680 nm). The obtained spectra were normalized to the setup sensitivity curve. PL decay kinetics were registered using an InGaAs/InP avalanche photodiode (MicroPhotonDevices, Bolzano, Italy) synchronized with 635 nm pulsed laser PicoQuant PDL 800-B with laser head LDH-P-C-635B (PicoQuant, Berlin, Germany). The PL decay excitation pulse energy was about 0.4 nJ, with pulse width < 90 ps and pulse repetition rate varied in the range of 0.5–1.0 MHz. Both steady-state and time-resolved measurements were taken from the area within ~80 μm^2^. To measure PL decay at 532 nm and 1064 nm wavelengths, a YAG: Nd laser with 1 MHz repetition rate and 10 ps pulse duration served as a radiation source. Acton grating monochromator and a single-photon receiver were used to record PL signal. The PL kinetics were recorded at the maxima of the luminescence bands corresponding to the emissions of InP NWs and InAsP QI and QW at the temperatures of 77 K and 300 K. The instrumental time resolution of the entire system was about 0.1 ns in the time-resolved PL measurements at all wavelengths. PL decay curves were fitted using exponential tailfit.

## 3. Results and Discussion

Figure 1a,b shows the luminescence spectra of the InAsP nanoinsertions in the absence of the TOPO ligand shell and with TOPO ligand shell for two concentrations (7 × 10^−7^ mol/L denote TL means TOPO with low concentration, and 1.5 mol/L denote TH means TOPO with high concentration) measured at 300 K. Note the characteristic features: the presence of a ligand shell leads to quenching of the QI luminescence (band 1.25–1.4 μm) with the increase of ligand concentration; at high concentration of the TOPO, the luminescence band belonging to the quantum well disappears (band 1.0–1.2 μm); and the spectrum itself undergoes a 200 cm^−1^ hypsochromic shift. The blue shift of the PL spectra of the nanowires is most likely associated with the Coulomb interactions of charges or dipoles distributed over the NW surface [7].

The surface passivation of nanocrystals (NCs) with such ligands as the TOPO can lead to an energetic rearrangement of existing trap states and/or to an emergence of new trap states near the energy band gap of the NC [11,12,13]. According to Fisher et al. [14], during the interaction of the ligand with the NC surface, three types of molecular orbitals (MOs) can emerge: MOs localized on the surface NC, MOs localized on the ligands, and the hybridized orbitals that are spread over both NC and ligand atoms. These hybridized states may enhance nonradiative relaxation from the band edge via electron-phonon coupling with high frequency vibrations of the ligands. On the other hand, it is known that a TOPO ligand, having double bonds, creates a metal-to-ligand covalent bond (e.g., in our case In-O-P) and creates new bonding/antibonding orbitals that are positioned either within or outside the intraband or interband energy gaps [15]. As a result, nonpassivated phosphorus atoms, which are hole acceptors, remain on the surface of InP NWs [16]. The latter can lead to an enhanced surface recombination and the emergence of an effective positive charge density on the surface of the NW. Thus, with an increase in the TOPO concentration, the PL quenching of QI and QW was observed as a result of a decrease in the energy supply of their emitting states from the InP NW array.

The deposition of the TOPO-CdSe/ZnS QDs layer on a InP/InAsP/InP NWs ensemble leads to a significant increase of PL lifetime and intensity of the InAsP QI [17], in contrast to the case of passivation with pure TOPO. Note that this effect is observed at all used excitation wavelengths: 532 nm, 635 nm and 1064 nm.

Table 1 shows the results of measuring the PL kinetics dependences for the emitting centers of QI and QW with different types of ligand shells at the excitation in the transparent region of QDs (635 nm).

The values presented in the Table 1 were calculated using the approximation by the function of the sum of two exponentials, which is widely known for describing various kinetic processes [9,18,19]:I_lum_ = A_1_ × exp(−t/t_1_) + A_2_ × exp(−t/t_2_), (1)
Here, I_lum_ is PL decay intensity normalized to unity at the initial moment of time; t_i_ is PL decay time components, and A_i_ is their corresponding amplitudes. The condition for the amplitudes is the next: A_1_ + A_2_ = 1. The use of this function is quite universal, and, as shown in [9], it makes it possible to characterize the radiating state by the number of main feeding channels. To calculate the average lifetime, we used the formula <t> = ∑A_i_ × t_i_, taking into account the presence of FRET in the system [8,9]. It can be seen from Table 1 that upon excitation at 635 nm wavelength, the kinetics of both InAsP QI and QW had two luminescence decay components with similar decay times: t_1_ = 4–6 ns and t_2_ = 25–30 ns (the average lifetime is about 10–12 ns). The faster component t_1_ corresponded to direct transition of the InAsP nanoinsertions, and the long-term t_2_ corresponded to the contributions of other channels of excitation energy exchange (population of excited states through electron-phonon and trapped states in the region of the InAsP/InP heterojunction, diffusion of electron-hole pairs (excitons) arising upon excitation of the InP NW shell, and processes at the NW-TOPO-QDs interface). Without passivation, an increase in temperature only led to a significant reduction in the long-term component t_2_ of the QW due to thermal exchange and, accordingly, in the average QW lifetime from 10 to 3 ns [9]. The deposition of low concentration TOPO led to an increase in t_2_ for QI and QW, which decreased with increasing temperature. QI kinetics had ceased to be thermostable. The deposition of high concentration TOPO also affected the exciton component t_1_ for both QI and QW. As noted above, the appearance of antibonding orbitals [13] at the InP-TOPO interface provoked the temporary capture of photoexcited carriers with their subsequent return to the exciton state (“reverse transfer”), which led to an increase in the lifetime of InAsP nanoinsertions. A similar process of transfer to traps and return of excitation energy with the participation of the TOPO ligand on CdSe/ZnS QDs was described in [20]. Indirect evidence of the formation of a new absorbing state is the emergence of a new luminescence band in the region of 700 nm (Figure 2) upon deposition of the TOPO-CdSe/ZnS QDs layer on the NWs and excitation of colloidal quantum dots in the absorption region. Note that such a band was absent during the passivation with pure TOPO.

The deposition of the TOPO layer (intermediate concentration in comparison with the considered ones) with CdSe/ZnS QDs on the NWs array led to an increase in the QI PL intensity by several times [8,17]. In this case, the relaxation times of QI and QW increased much more than in the case of pure TOPO deposition (Table 1). As the temperature rose, the deposition of QDs prevented the quenching of the QW luminescence. With such passivation, another reason for the increase in the lifetime of InAsP nanoinsertions is a different morphology of the layer that appeared at the NW ligands interface as a result of the other orientation of the TOPO molecular bonds. The latter and the charge redistribution on the NW surface together led to an energetic rearrangement of low-energy trapped states, both surface states and states at the interfaces between QI, QW, and the InP NW volume.

For the case of 532 nm excitation the InAsP QI, PL itself had a higher intensity and a narrower band compared with the case excited by radiation of 635 nm wavelength at the same pump power [8]. This trend continued during the deposition of the TOPO-CdSe/ZnS QDs layer. Note that the wavelength of 532 nm was in the absorption region of CdSe/ZnS QDs, but the wavelength of 635 nm was not. At low temperatures, the TOPO-CdSe/ZnS QDs layer had no effect on the spectral characteristics of NWs excluding PL increase (Figure 3).

The luminescence band in the 1.25–1.4 µm region belongs to the emission of the InAsP QI, and the shorter wavelength band of 1.0–1.2 µm is associated with the emission of the radial QW [10]. At 77 K, a low-intensity band was observed in the region of 880 nm, corresponding to InP emission. An increase in temperature changed the spectral picture: the QI and QW spectra merged into a wide spectrum with a noticeable bathochromic shift. A noticeable difference was also present in the results of the luminescence kinetics. Table 2 shows a comparison of the PL decay components, calculated by function (1), of the InP/InAsP/InP NWs heterostructure with and without the deposited CdSe/ZnS-TOPO QDs layer under excitation at 532 nm wavelength and at two temperatures. The initial InAsP QI decays monoexponentially with the natural lifetime of about 13 ns [9]. The QW kinetics had two components, 3 and 23 ns at 77 K with the same contributions, and an average lifetime of about 13 ns. It was shown in [9] that there is energy transfer from a QW to a QI upon excitation at a wavelength of 635 nm. The monoexponentiality of InAsP QI kinetics upon excitation of 532 nm and its dependence on temperature, in contrast to the excitation at 635 nm wavelength, proved the dominance of direct population of QI excited states.

The presence of the TOPO-CdSe/ZnS QDs layer led to a lengthening of the average relaxation times, which were much shorter in comparison with the case of excitation at 635 nm wavelength. At low temperatures, the participation of CdSe/ZnS QDs at the ligand shell was reduced to passivating effect and direct radiative transfer to the QI. The Förster mechanism was also possible with CdSe/ZnS quantum dots acting as energy donors and QW and QI acting as acceptors [8]. Temperature increase led to the manifestation of both components of the QI luminescence in this case, which was probably caused by the intensification of low-energy traps deactivation and the “reverse transfer” of energy from them to the QI.

To exclude the excitation of both colloidal QDs and the InP NW, spectroscopic studies of the NWs were carried out at 1064 nm wavelength. Comparison of the spectra on Figure 4 shows that the presence of the TOPO-CdSe/ZnS QDs shell led to several folds increase in intensity. At the same time, the ratio of PL intensities of the samples is approximately the same for both pumping at 532 nm wavelength and at 1064 nm.

In this case, the PL kinetics of InAsP nanoinsertions had short times: less than 1 ns for the initial NWs and 5 ns after passivation with the TOPO-CdSe/ZnS QDs layer. The decay times were shorter because of no excitation energy was transferred from the InP array to QI and QW, in contrast with the case of excitation with a 635 nm wavelength [9]. We suppose that an increase in PL intensity and lifetime on TOPO-CdSe/ZnS QDs passivation is associated with the increase in the probability of excitation transfer from QW to QI due to the decrease in the barrier between them and, probably, the decrease in the number of trap states.

## 4. Conclusions

In summary, we have investigated the influence of the ligand coverage of TOPO and quasi-Langmuir layer of TOPO-CdSe/ZnS QDs on PL dynamics of the InP/InAsP/InP NWs ensemble. First of all, we note the role of the TOPO-CdSe/ZnS QDs layer as a novel effective surface passivation coating for NWs. The spectroscopic measurements showed that the presence of the TOPO ligand led to a hypsochromic shift of the PL maximum, an increase in the decay time, and a significant quenching of the luminescence. The PL quenching increased with the density of the ligand coverage. In turn, the deposition of the TOPO-CdSe/ZnSQDs layer led to an increase in the luminescence intensity of the InAsP nanoinsertions by an order of magnitude on average at all the excitation wavelengths used in the work. The decay times for TOPO-QDs coating exceeded the values for TOPO coating (with 635 nm wavelength excitation). The results showed that the localization of electrons on anti-bonding orbitals and hybridized states played a dominant role in the increase of PL lifetime. The reason for the above-mentioned differences is the different morphology of the emerging layer at the NW-ligand interface for the two cases, which led, not only to different passivation of the surface states, but also to a different transformation of the NW surface, to a charge redistribution, and, accordingly, to the energy rearrangement of low-energy trap states on both surface and states at the interface between QI and QW and the InP NW volume. It is possible that a change in the ligand coverage led to a change in the energy state of the heterojunction between the InAsP nanoinsertions and the InP NW volume. This actually changes the general idea of the effect of low-energy traps on the luminescence intensity since the intrinsic quenching of excited states associated with the capture of excitation on the traps is compensated by the increasing role of the “reverse transfer”. Thus, the existence of several competing channels of nonradiative and radiative recombination determines the nontrivial picture of the PL dynamics of the hybrid nanostructure InP/InAsP/InP NWs-TOPO-CdSe/ZnS QDs described above. The TOPO-CdSe/ZnS QDs can be very effective as a novel surface passivation coating for NWs.

## Figures and Tables

**Figure 1 nanomaterials-11-00640-f001:**
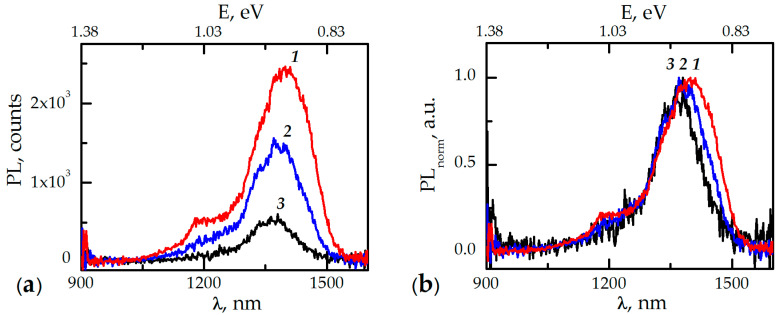
The absolute (**a**) and normalized (**b**) photoluminescence (PL) spectra of the InAsPnanoinsertions measured at 300 K: ***1*** is in the absence of the trioctylphosphine oxide (TOPO) ligand shell; ***2*** is for 7 × 10^−7^ mol/L TOPO concentration; and ***3*** is for 1.5 mol/L TOPO concentration.

**Figure 2 nanomaterials-11-00640-f002:**
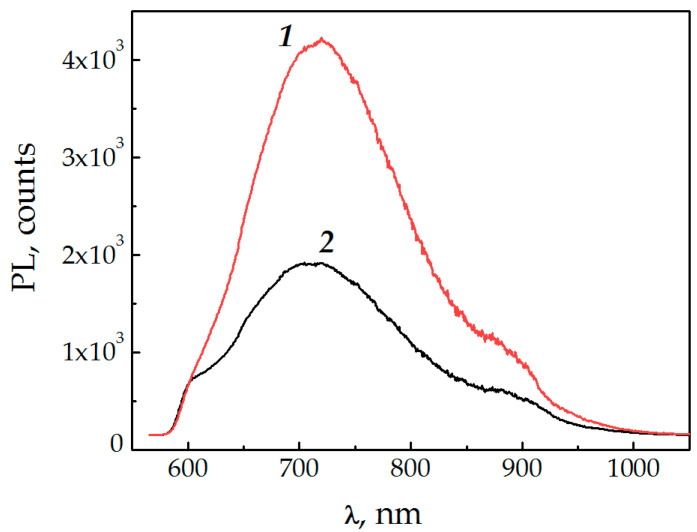
PL spectra of the InP/InAsP/InP-TOPO-CdSe/ZnS heterostructure at 532 nm wavelength excitation: ***1*** at T = 77 K; ***2*** at 300 K.

**Figure 3 nanomaterials-11-00640-f003:**
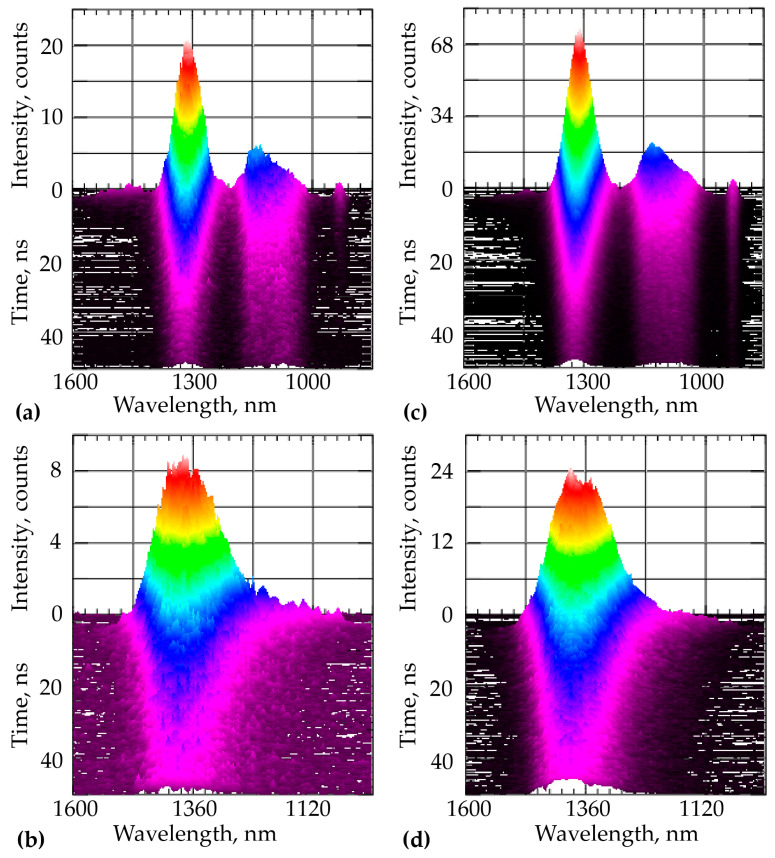
PL spectra of the heterostructures at 532 nm wavelength excitation: (**a**) nanowires (NWs) at T = 77 K; (**b**) NWs at T = 300 K; (**c**) NWs with the TOPO-CdSe/ZnS QDs layers at T = 77 K; and (**d**) NWs with the TOPO-CdSe/ZnS QDs layers at T = 300 K.

**Figure 4 nanomaterials-11-00640-f004:**
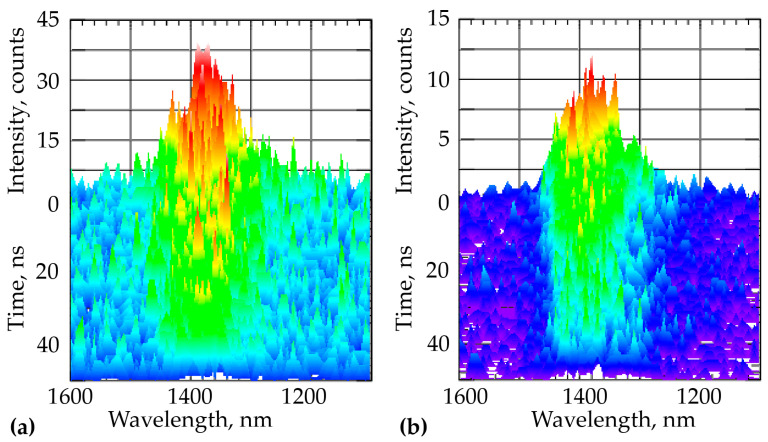
PL spectra of the InP/InAsP/InP NWs heterostructure: (**a**) at 1064 nm wavelength excitation and at T = 77 K: ***A*** is NWs; (**b**) is NWs with the TOPO-CdSe/ZnS QDs layers.

**Table 1 nanomaterials-11-00640-t001:** The parameters of nanoinsertion (QI) and quantum well (QW) kinetics at 635 nm wavelength excitation.

Kinetic Parameters	QI (InAsP)		QI-TL		QI-TH		QI-TOPO-QDs(CdSe/ZnS)	
	77–300 K		77 K	300 K	77 K	300 K	77 K	300 K
**A1**	0.7		0.80	0.87	0.67	0.65	0.65	0.57
**t1, ns**	6		8	4	14	8	28	14
**A2**	0.3		0.20	0.13	0.33	0.35	0.35	0.43
**t2, ns**	25		52	32	75	57	125	78
**<t>, ns**	12		17	8	34	25	62	42
	**QW (InAsP)**		**QW-TL**		**QW-TH**		**QW-TOPO-QDs(CdSe/ZnS)**	
	**77 K**	**300 K**	**77 K**	**300 K**	**77 K**	**300 K**	**77 K**	**300 K**
**A1**	0.76	0.60	0.75	0.60	0.55	x	0.57	0.60
**t1, ns**	4	5	4	2	8	x	9	4
**A2**	0.24	0.40	0.25	0.40	0.45	x	0.43	0.40
**t2, ns**	30	1	37	9	45	x	47	27
**<t>, ns**	10	3	12	5	25	extinguished	25	13

**Table 2 nanomaterials-11-00640-t002:** The parameters of QI and QW kinetics at 532 nm wavelength excitation.

Kinetic Parameters	QI (InAsP)		QI-TOPO-QDs(CdSe/ZnS)	
	77 K	300 K	77 K	300 K
**A1**	1	1	1	0.75
**t1, ns**	12.6	8.7	16	6.9
**A2**	-	-	-	0.25
**t2, ns**	-	-	-	54.8
**<t>, ns**	-	-	-	18.9
	**QW (InAsP)**		**QW-TOPO-QDs(CdSe/ZnS)**	
	**77 K**	**300 K**	**77 K**	**300 K**
**A1**	0.5	1	0.67	x
**t1, ns**	2.8	2.2	6.6	x
**A2**	0.5	-	0.33	x
**t2, ns**	22.6	-	39.8	x
**<t>, ns**	12.7	-	17.6	extinguished

## Data Availability

The data presented in this study are available on request from the corresponding author. The data are not publicly available due to privacy.

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
