# Peer review of "Influence of TOPO and TOPO-CdSe/ZnS Quantum Dots on Luminescence Photodynamics of InP/InAsP/InPHeterostructure Nanowires"

_nanomaterials, 2021, doi:10.3390/nano11030640_

Round 1

Reviewer 1 Report

Manuscript ID:  nanomaterials-1123048
Title:     Influence of TOPO and TOPO-CdSe/ZnS quantum dots on luminescence 
        photodynamics of InP/InAsP/InP heterostructure nanowires
Authors:     Artem Khrebtov, Vladimir Danilov, Anastasia Kulagina, Rodion 
        Reznik, Ivan Skurlov, Alexander Litvin, Farrukh Safin, Vladislav Gridchin, 
        Dmitriy Shevchuk, Stanislav Shmakov, Artem Yablonskiy, George Cirlin

The authors report experimental results on the luminescene of an InP/InAsP/InP heterostructure. Specifically, they investigate the influence of a passivation layer, i.e. troctylphosphine oxide (TOPO) with / without additional CdSe/ZnS quantum dots, on the photodynamics of the heterostructure. The photoluminescence is reported after excitation with three different wavelengths. The luminescene spectrum of two different InAsP structures within the InP nanowire are observed at wavelength between 1000 nm to 1400 nm.
The passivation layer quenches luminescene but TOPO with quantum dots a lot less than without. Using TOPO with quantum dots exhibits also additional luminescene around 700 nm. Beyond the spectra intensities the authors studied the kinetics. They find that the TOPO layer increases the life times of the luminescene substantially. The findings are rationalized by model assumptions regarding localized electrons and low-energy traps.

Semiconductor nanowires are of great potential use for future solar cells and photodetectors or emitters. Reliable, stable and effective surface passivation is needed. Thus, the influence of these passivation layers on the optical performance is of central importance. Thus, the results are interesting and the paper is clearly written. Before I can recommend publication I would like the authors to address the following points:

1. The authors report that the quenching by a passivation layer of TOPO including the quantum dots is substantially less compared with a TOPO layer without quantum dots. There is, however, no plot of according experimental results. Obviously, an according line for the PL spectra with TOPO including the quantum dots in Figure 1 would be enlightning. 

2. The various results specifically of the change in kinetic life times are rationalized with various model assumptions mostly without clearly citing according published results. These descriptions must be supported by references.

3. All used model descriptions are in some sense 'microscopic' discussing orbitals and bonds and so on. Energetic shifts and also changes of life times due to additional layers can also be understood by simple models taking only an average dielectric behaviour of the layer into account. Such model descriptions follow Onsager like pictures. See, for example,  JCP 141, 044304 (2014).

Author Response

We thank the reviewer for carefully reading the work and making valuable comments.

Response 1: Figure 1 shows the PL comparison spectra of NWs without and with the TOPO passivating layer for two different concentrations. It follows from them that an increase in the TOPO concentration leads to more quenching. The TOPO layer containing CdSe/ZnS QDs acts opposite to the pure TOPO layer, namely, increases the PL amplitude of NWs, and at all excitation wavelengths. With excitation at 532 nm wavelength, see Fig. 3 (a), 3 (c), at 1064 nm excitation see Fig. 4, at 635 nm excitation see Semiconductors 2019 V. 53(9) p1289, which we cite in the paper.

Response 2: Mathematically, we have used just one model for approximation of all kinetic experimental results, namely the two exponential decay function which is mostly used in common description. Some references proving the use of this function are already given. At the request of the reviewer, we have added citation of important papers giving the interpretation of kinetic parameters.

Response 3: The example proposed by the reviewer does not at all prove the simplicity of using the Onsanger-Betcher model in comparison with those proposed in the authors' work, but suggests an alternative approach, which the authors propose to implement in the future.

Reviewer 2 Report

Authors report on the influence of TOPO and TOPO-CdSe/ZnS quantum dots (QDs) on the optical properties of InAsP quantum insertions (QIs) embedded in InP nanowires. Photoluminescence (PL) and time-resolved PL studies are performed at room and 77K using lasers with different excitation wavelengths to demonstrate that a TOPO-CdSe/ZnS QDs shell can increase the PL intensity and the decay time of the InAsP QIs. I think that the results worth publishing. However, I am not convinced by some claims of the work and I have the following comments:

1) "But still, there is no clearly established, stable and effective surface passivation method for the InP / InAsP / InP heterostructured NWs." I don't really understand this sentence. To obtain efficient emission, the quantum dot (QD) or insertion (QI) must be inserted into a 300 to 350 nm diameter nanowire (NW) in order to guide efficiently the telecom emission from the QD (for instance: Haffouz et al., Ref [4]). Therefore, all the work performed on the passivation of InP NWs, seems compatible and directly transferrable to InP NWs containing InAsP QDs or QIs.

2) Photoluminescence (PL) spectra (Fig 1) reveal a decrease of the PL emission of the QIs when the nanowires (NWs) are covered by a TOPO ligand shell. The authors explain that the TOPO ligand shell leads “to an enhanced surface recombination”. This seems to contradict the results of Van Vugt et al., where the PL emission efficiency of the InP NWs passivated by a TOPO shell was better than unpassivated ones. How do you explain these different results? Have you measured the PL signal from InP NWs with and without TOPO?

3) Are the time-resolved PL measurements deconvoluted by the experimental response of the experimental system? What is the instrumental time resolution of the entire system?

4) Fig 2: Why can't we see the emission peak of the Wz InP? Which detector is used to measure the signal in this wavelength range? The “Materials and Methods” paragraph speaks of an InGaAs detector but for Fig 2 an Si detector would be more relevant.

5) I think that the quality of Figures 3 and 4 should be improved.

6) Why are the dynamics of the QIs not the same for an excitation of 635 nm and 532 nm (for instance, the reference sample without the TOPO layer in tables 1 and 2)? In both cases, the excitation is above the bang gap of Wz InP. The same phenomenon is observed for the dynamics of the radial QW.

7) I have the same question with Figure 4. Excitation of the QIs with a 1064 nm laser  shows a longer lifetime when the NWs are covered with the TOPO-CdSe/ZnS QD layer, which confirms the passivation hypothesis. However, the measured lifetime on the reference sample is very different from values obtained with an excitation above the InP band gap. Can the authors explain these different results?

Author Response

We thank the reviewer for carefully reading the work and making valuable comments.

Response 1: Definitely, we agree with the remark. We delete the sentence, which was not clear for the reviewer. The reviewer absolutely right saying that the increasing of the shell thickness will lead to the decreasing of the surface states of the NWs, and, additionally may provide a guide effect. However, this is not simple task during MBE growth and alternative shells which give an increase of PL intensity is a challenging task. Indeed, InP NWs passivation methods are applicable to our heterostructured NWs as well.

Response 2: Vugt et. al. have used a different TOPO surface passivation method, where PL efficiency of InP NWs was improved by photoassisted wet chemical etching in a butanol solution containing HF and TOPO. In our case, we did not use HF. 

Response 3: The instrumental time resolution of the entire system was about 0.1 ns in the time-resolved PL measurements at all wavelengths. We added it in the text. This resolution was much shorter than the typical PL decay times obtained in the experiment (a few nanoseconds or longer). Therefore, the deconvolution of the obtained PL decay curves was not performed. 

Response 4: In fact, we observed a week emission at ~ 900 nm on Fig.2, which can be attributed to the emission from InP NWs. PL spectra presented in figure 2 were indeed measured using Si-based detector. We have corrected the Materials and Methods section.

Response 5: We have improved the quality of Figures 3 and 4.

Response 6: In fact, the difference in the lifetimes of original InAsP QI and QW is not very pronounced (e.g. for 77K: 12/10 ns from table1 and 12.6/12.7 ns from the table2) and it can be just experimental peculiarities (different PL kits and little inhomogeneity over the sample).

Response 7: We should emphasize that the excitation at 1064 nm is closer to the absorption band of nanoinsertions (quasi resonant excitation) and we did not excite the InP NW. The decay times are shorter in this case because of no transfer of excitation energy from the InP array to QI and QW, in contrast to the case of excitation with 635 nm wavelength. We have added a corresponding explanation in the text.

Round 2

Reviewer 1 Report

The authors have addressed my previously raised concerns with a minimalistic response. I feel nevertheless that I can recommend publication.

Reviewer 2 Report

The authors have answered all my comments, improved and clarified the draft . Therefore, I recommend the publication of this work in nanomaterials.